# Isomeric Activity Cliffs—A Case Study for Fluorine Substitution of Aminergic G Protein-Coupled Receptor Ligands

**DOI:** 10.3390/molecules28020490

**Published:** 2023-01-04

**Authors:** Wojciech Pietruś, Rafał Kurczab, Dawid Warszycki, Andrzej J. Bojarski, Jürgen Bajorath

**Affiliations:** 1Department of Medicinal Chemistry, Maj Institute of Pharmacology, Polish Academy of Sciences, Smetna 12, 31-343 Krakow, Poland; 2Department of Life Science Informatics, LIMES Program Unit Chemical Biology and Medicinal Chemistry, B-IT, Rheinische Friedrich-Wilhelms-Universität, Friedrich-Hirzebruch-Allee 6, D-53115 Bonn, Germany

**Keywords:** G protein-coupled receptor, GPCR, fluorine, activity cliffs, AC, MMP, ChEMBL, induced fit docking, MD, QPLD

## Abstract

Currently, G protein-coupled receptors (GPCRs) constitute a significant group of membrane-bound receptors representing more than 30% of therapeutic targets. Fluorine is commonly used in designing highly active biological compounds, as evidenced by the steadily increasing number of drugs by the Food and Drug Administration (FDA). Herein, we identified and analyzed 898 target-based F-containing isomeric analog sets for SAR analysis in the ChEMBL database—F_i_SAR sets active against 33 different aminergic GPCRs comprising a total of 2163 fluorinated (1201 unique) compounds. We found 30 F_i_SAR sets contain activity cliffs (ACs), defined as pairs of structurally similar compounds showing significant differences in affinity (≥50-fold change), where the change of fluorine position may lead up to a 1300-fold change in potency. The analysis of matched molecular pair (MMP) networks indicated that the fluorination of aromatic rings showed no clear trend toward a positive or negative effect on affinity. Additionally, we propose an in silico workflow (including induced-fit docking, molecular dynamics, quantum polarized ligand docking, and binding free energy calculations based on the Generalized-Born Surface-Area (GBSA) model) to score the fluorine positions in the molecule.

## 1. Introduction

G protein-coupled receptors (GPCRs) constitute a significant group of membrane-bound receptors that contain five different classes [1] and represent more than 30% of therapeutic targets [2]. The aminergic subfamily of receptors is present in class A and includes receptors for acetylcholine, dopamine, epinephrine, histamine, serotonin, and trace amine [1]. Class A aminergic GPCRs are the most vastly studied and drugged subfamily of GPCRs [3], which are used in the treatment of neurodegenerative, immunological, cardiac, and renal diseases. They are also used in treating cancer and many other disorders, as well as in many cases acting as important off-targets [4].

Fluorine is a commonly used substituent in medicinal chemistry, which has resulted in a large number of fluorinated, marketed drugs [5]. Fluorine-containing compounds constitute over 50% of blockbuster drugs [6], e.g., (aggregated sales for 2021) Biktarvy (USD 8.624 Billion), Trikafta (USD 5.697 Billion), Xtandi (USD 5.636 Billion) and Invega (USD 4.022 Billion). In the last decade (2011–2020), fluorinated drugs constituted 30% of all compounds approved by the Food and Drug Administration (FDA) [7]. Additionally, fluorinated drug candidates that enter clinical trials are frequently observed (called emerging drugs), e.g., GS-6207 (lenacapavir—III phase, HIV-1) [8], AZD-9833 (camizestrant—III phase, breast cancer) [9] or BI-425809 (iclepertin—III phase, schizophrenia) [10]. Fluorine has a significant impact on bioavailability, lipophilicity, metabolic stability, acidity/basicity, and toxicity [11]. Because of the complex effect of fluorination, rules of thumb for estimating the effects of fluorine substitution still do not exist [4].

Herein, we examined fluorine isomers of compounds in the context of activity cliffs (ACs), which are generally defined as pairs of structurally similar compounds having a large difference in potency. Our previous results, which included pairs of unfluorinated compounds and their fluorinated analogs, suggested that fluorination in the ortho position of aromatic rings led to an increase in potency, whereas fluorination of aliphatic fragments more often led to a decrease in biological activity [4].

The accurate prediction of protein−ligand binding energy becomes a central task in computer-aided drug design; however, standard scoring functions do not take into account complex inductive or resonance effects. Because of the highest electronegativity of fluorine and its effect on the electron density distribution of adjacent groups/atoms, docking and scoring seem to be insufficient for the evaluation and selection of new fluorinated derivatives for synthesis. Thus, we also proposed an in silico workflow (which includes induced-fit docking (IFD), molecular dynamics (MD), quantum polarized ligand docking (QPLD), and binding free energy calculations based on the Generalized-Born Surface-Area (GBSA) model) to compensate for limitations of standard scoring functions and increase the accuracy of predicted poses and docking scores, as well as improve the discrimination of active from inactive compounds. This workflow was validated based on experimental data F_i_SAR sets containing ACs, assessing the possibility of correlating biological activity data with energy changes calculated in the GBSA model.

## 2. Results

### 2.1. F_i_SAR Sets

All high-confidence ligands of 35 GPCR targets were extracted from ChEMBL. A search for fluorinated sets led to the assembly of 898 target-based F_i_SAR sets with activity against 33 GPCRs, comprising a total of 2163 fluorinated (1201 unique) compounds. If two or more fluorinated compounds shared the same chemical core (fluoro isomers) and had the same target annotation, they were grouped into one target-based F_i_SAR set, as shown in Table 1. On this basis, 898 target-based F_i_SAR sets were identified containing 2163 fluorinated compounds. The size of F_i_SAR sets ranged from two to seven isomers (603 sets–2 cmpds, 238–3, 48–4, 5–7, 6–1, 7–1) with experimental activity values for 33 individual targets. The largest number of fluorinated compounds was found for the dopamine D2 receptor (284 compounds in 112 sets) and serotonin receptors 5-HT1a, 5-HT2a, 5-HT6, and 5-HT7 (255–102, 230–97, 176–75, and 156–72 sets, respectively). For each F_i_SAR set, the potency difference within the target was determined. More than half (~64%) of the F_i_SAR sets showed a positive potency trend for the target caused by different fluorine isomers, and the rest of the sets showed no significant potency change (ΔpPot ≤ 0.3, which indicates 2-fold differences).

### 2.2. Activity Cliffs in F_i_SAR Sets

The individual potency changes (expressed as |ΔpPot|) for compounds the in F_i_SAR sets in ranged from 8.45 to 0.0067 on the logarithmic scale. Among all F_i_SAR sets, compounds in 149 sets displayed at least a 10-fold change, and 47 sets displayed a 30-fold improvement in biological activity. However, only 30 F_i_SAR sets contained compounds with |ΔpPot| greater than 50-fold (ΔpPot ≥ 1.7), classified as ACs. Whereas the potency difference criterion for standard MMP-cliffs is often set to 100-fold (ΔpPot ≥ 2.0), we modified this potency difference criterion since we only considered F-isomers (a very small change in the structure). Exemplary ACs with different potency values are illustrated in Table 1.

In the first example, the change in the fluorine atom position of muscarinic M_1_ receptor ligands had a significant effect on biological activity. In the original publication, no explanation of the role of fluorine in the 3-chlorophenyl ring was found [12]; however, the authors proposed that fluorine in ortho position to an amide fragment creates an intramolecular hydrogen bond (iHB) with amide hydrogen causing stabilization of a favorable conformation [13]. Nevertheless, the authors attempted to rigidify this conformation by cyclization of this fragment, but this did not yield the desired results. The compounds were found to be inactive [13]. Due to the lack of in silico studies, it can only be suspected that the effect of fluorine on the donor properties of the halogen bond of the chlorine atom and the amine group had a significant effect on the biological activity.

In the second example of the serotonin 5HT2a receptor (Table 1, the authors indicated that aryl groups bearing electron-withdrawing groups (F) tended to retain good potency, whereas compounds possessing electron-donating groups (e.g., OMe) tended to be less potent [14]. Additionally, the authors showed that the position of fluorine influenced the selectivity of compounds toward 5-HT2a, 5-HT2b, and 5-HT2c receptors. It is worth noting that the 4-fluoro derivative showed the highest activity toward the 5-HT2a receptor (EC_50_ = 23 nM, 3 nM and 648 nM for 5-HT2c, 5-HT2a, and 5-HT2b, respectively), 3-fluoro toward 5-HT2b (EC_50_ = 25 nM, 32 nM and 13 nM for 5-HT2c, 5-HT2a, and 5-HT2b, respectively), and 2-fluoro toward 5-HT2c (EC_50_ = 5 nM, 648 nM, and 94 nM for 5-HT2c, 5-HT2a, and 5-HT2b, respectively), and at the same time, this derivative was the most selective one. These results clearly show that one fluorine atom can be used for tuning the selectivity and biological activity toward even very similar targets from one subfamily [14].

### 2.3. MMP Networks

To further extend the SAR information contained in the F_i_SAR sets, a previously designed variant of an MMP network with multiple information layers was used [4]. Hence, the 898 F_i_SAR target-based sets were combined into a single, multitargeted F_i_SAR set if they shared the same chemical core and differed only in fluorine position, as schematically shown in Table 2. For 263 of the target-based F_i_SARs, no further F_i_SAR set was found that shared the same chemical core, and they remained as single-target sets. These 177 combined F_i_SAR sets consisted of compounds with multiple target annotations.

The resulting 177 F_i_SAR sets were used for MMP network generation, where each node corresponded to one of these sets. The 89 single-target sets are represented as circles, the double-targeted (89 sets) as squares, and the multitargeted (≥3 targets) are represented as rhombuses. Two nodes were connected by an edge if they were structurally similar (the similarity was calculated between nonfluorinated compounds identified by MMP among the F_i_SAR sets). The nodes are color-coded according to the predominant subfamily in the analyzed multitargeted F_i_SAR sets (cyan: serotoninergic, navy: dopaminergic, green: adrenergic, magenta: histaminic, yellow: muscarinic). In addition, a thick border is drawn around the nodes if ACs were found in the underlying F_i_SAR set. An exemplary network is depicted in Figure 1.

The exemplary F_i_SAR set (Figure 2) containing four different aromatic substituents (pyrimidine, oxazole, and two isomers of pyrazole) in the ortho position to the 5H,6H,7H-pyrrolo [1,2-a]imidazole fragment consisted of partial agonists of the alpha-1a adrenergic receptor. The authors used fluorine to gain selectivity among all subtypes of adrenergic receptors toward the alpha-1a receptor [15]. Additionally, fluorine reduced E_max_ and modulated HLM stability and flux in the MDCK assay. Placement of fluorine at the *meta* position (to the bicyclo fragment) consistently lowered the E_max_ across a range of heterocycles in the *para* position and resulted in selectivity. However, the authors did not investigate the effect of the position of fluorine and its substitution preference. Fluorine substitution can lead to an important change in the basicity of the imidazole fragment and the distribution of the electron density of 5H,6H,7H-pyrrolo [1,2-a]imidazole.

### 2.4. Computational Workflow to Rank the Positions for Fluorine Substitution

Notably, the introduction of fluorine to bioactive compounds can lead to either an improvement or a deterioration of biological activity [4,7,16,17]. However, rules of thumb for fluorination to obtain highly potent compounds have not been defined thus far. Designing new drugs typically involves the use of modern computational techniques such as molecular modeling, machine learning, and quantum-mechanical calculations. Simple docking applications are defined by a set of rules and parameters applied to predict the conformations and the ranking of the designed compounds using scoring functions. However, because of the highest electronegativity of fluorine among all elements, these scoring functions do not take into account inductive or resonant effects and the influence of fluorine on the electron density distribution of adjacent groups/atoms. Therefore, these methods seem insufficient for the evaluation and selection of new fluorinated derivatives for synthesis.

We selected F_i_SAR sets with ACs for the dopamine D2 receptor (PDB ID: 6CM4), serotonin 5-HT1a receptor (PDB ID: 7E2Z), serotonin 5-HT2a receptor (with MMP connection) (PDB ID: 6A93) and muscarinic M1 (PDB ID: 5CXV) and M2 receptors (PDB ID: 5ZHP). The receptor conformation during the standard docking of the new ligand was rigid; thus, the first step in our algorithm was to dock a nonfluorinated compound (which is not present in the F_i_SAR sets) using the induced-fit docking (IFD) algorithm, which can propose new conformationally adjusted ligand binding modes and induce structural changes in the receptor. Fluorine is a bioisostere of hydrogen, so it should not cause significant conformational changes or the orientation of the molecule in the binding pocket. For each selected F_i_SAR set, the binding mode was selected corresponding to the literature reports (e.g., for aminergic GPCRs, one of the most important interactions was retaining the salt bridge with the D3.32 residue). For the obtained L–R complexes, 100 ns-long molecular dynamics simulations were performed to adjust the conformation of the entire binding pocket space to fit the ligand and highlight the most important interactions stabilizing a given system. Then, to determine the most common protein conformation, clustering of the obtained MD trajectory was performed based on the RMSD backbone matrix of each receptor, which was then used to generate the grid used in the next steps. It is emphasized that the QPLD approach was used to replace the fixed charges of ligands obtained from force field parameterization with the values calculated using QM/MM in the protein environment on the prediction of ligand–receptor complexes caused by the introduction of fluorine. In the next step, the RMSD was calculated between all fluorine derivative poses and the core conformation complex obtained after clustering of the MD trajectory. MM/GBSA was used to calculate the binding free energy based on the receptor-ligand complexes for three poses obtained at the QPLD stage with the smallest RMSD to the core. Then, we estimated the energy change of the fluorinated derivatives compared to the most active F-analog in a given F_i_SAR set.

In the first series of 5-HT1a ligands (Figure 3), the position of fluorine in the quinoline ring significantly affects the affinity and metabolic stability [18]. The C–F bond is highly polarized, and it gains great stability caused by the electrostatic attraction between C^δ+^ and F^δ−^ atoms [19], making fluorine a good inhibitor of potential hydroxylation sites [11]. Substituting the 5-position with a fluorine group retained potent 5-HT1a full antagonist activity; however, substitutions in the 6- and 7-positions deteriorated the intrinsic activity in the in vitro cAMP assay. According to the authors, this suggested a significant influence of fluorine on the binding mode or electron density distribution [18] (Figure 3). Substitution of fluorine in the 3-position decreased intrinsic activity; however, increases in stability were observed in human microsomal preparations [19]. Notably, in the analyzed binding modes obtained in the QPLD approach, fluorine did not interact with any amino acids because the quinoline ring was directed outside the receptor. The exception was the 3-fluoro derivative, where fluorine interacted with Q2.65. It is worth emphasizing that the QPLD approach calculates the charges on atoms using quantum methods, so it takes into account resonance and inductive effects. The quinoline ring interacted with Y2.64, N7.39, and W7.40; therefore, the aromaticity of the conjugated aromatic system significantly affected the activity of the compounds. The ΔΔG values resulting from our workflow closely correlated with biological data (ΔpK_i_) (Figure 3), which also suggests that fluorine does not create its own stabilizing effects. However, the induced effects are an important factor influencing the potency change [7].

The next F_i_SAR example (Figure 4) illustrates the 5-HT2a ligands with a difference in the phenethanone piperazine linker or phenethylpiperazine linker [20]. Switching the fluorine position from *para* to ortho in phenethylpiperazine linker derivatives slightly decreased the binding affinity, but a significant loss was observed with the fluoro group at the *meta* position (Figure 4). A similar trend was also observed in phenethanone piperazine linker derivatives because the ortho-fluoro isomer had a weaker 5-HT2a binding affinity than the *para*-fluoro isomer. The authors suggested that *para*-substitutions were most favored, followed by ortho-substitutions, whereas *meta*-substitutions were not well tolerated. Analysis of the binding mode in the 5-HT2a crystal showed that the phenyl ring interacts with neighboring aromatic amino acids, such as W6.48 and F6.52, and that only the *para*-fluoro derivative did not cause steric hindrance or clashing with those amino acids. Additionally, fluorine in the *para* position symmetrically shifted the electron density toward itself, increasing the acidity of the protons and influencing π-π interactions. In both cores, our workflow correctly predicted the structure–activity relationship (Figure 4).

Another F_i_SAR set was originally composed of multi-receptor compounds, but we focused on their action as antagonists of the dopamine D2 receptor [21]. In this study, the authors adopted a standard trial and error strategy to introduce halogens and a methyl group at possible substitution sites and to test their effect on activity/selectivity toward specific biological targets (i.e., D2, serotonin 5-HT1a, 5-HT2a and serotonin transporter (SERT)). The authors aimed to obtain good pharmacokinetic properties and the desired ratio of biological potency but did not explain the influence of fluorine in the binding mode. The analysis of the docked compound showed that fluorine in the 5-position did not interact with any amino acids in the binding pocket; however, switching fluorine to the 7-position deteriorated the potency, which could be caused by the electrostatic changes in the amide fragment (Figure 5). The substitution of fluorine in the quinoline ring, which is strongly engaged in π-π interactions with F5.47, caused decreases in biological activity. In this case, the workflow of quantum-polarized ligand docking to the prepared conformation of the protein also correlated well with the corresponding energy loss of the decreasing potency value (Figure 5).

Herein, the authors suggested that fluorine in the ortho position of the amide fragment created an intramolecular hydrogen bond with the amide hydrogen, causing stabilization of a favorable conformation [12,13]; however, in the obtained poses, fluorine did not form an iHB (Figure 6). The analysis of binding mode showed that chlorine formed a halogen bond with Y3.33; however, fluorine (which is a well-known atom that increases the sigma hole of adjacent halogens) caused a decrease in biological activity compared to the most active isomer. Additionally, switching fluorine from a 2-position (in comparison to amide) to a 3- or 5-position also deteriorated the potency, suggesting that the influence of fluorine on the carbonyl oxygen, which is engaged in hydrogen bonds with Y3.33 and Y45.51, may be crucial in the proposed binding mode (Figure 6). In our work, the charges were computed ab initio, which allowed for determining the influence of the inductive and resonanance effects in the molecule on atomic charges. By contrast, in standard docking studies, charges are frozen. The results obtained with our workflow appropriately ranked the compounds based on ΔΔG concerning the experimental values of biological activity (Figure 6).

The last example involves ligands of the muscarinic M3 receptor [22]. The authors conducted an exploration of multiple regions of a biaryl amine using fluorine, chlorine, methyl, and other standard substituents used in medicinal chemistry. The authors did not explain the preferred position of fluorine due to the lack of docking studies. Our analysis of binding modes showed that fluorine in the *para* position to the amine fragment (Figure 7) is engaged in HB with N6.52. In another position, fluorine did not interact with any amino acids, but the phenyl ring was located in the neighborhood of aromatic amino acids, which implied that π-π interactions might have an important role in stabilizing L–R complexes. Because of the high electronegativity of fluorine, the fluorination of the aromatic ring linked to piperazine might affect the basicity of the nitrogen atom, which was involved in the salt bridge with D3.32. The workflow correctly predicted ΔΔG between all fluoro isomers and correlated well with differences in biological activities (Figure 7).

The F_i_SAR sets with ACs showed that fluorine had substantial effects on biological activity. The correlation between experimental values and predicted losses (gains) in interaction energy of L–R complexes can be studied using the proposed in silico workflow. It is worth emphasizing that the preparation of the appropriate conformation of the protein for the molecular core of interest can be time-consuming and requires a more precise analysis of the binding mode, and the performance of molecular dynamics with appropriate parameters, taking the membrane environment into account. However, it likely allows for a better prediction and design of new fluorine or halogen derivatives based on theoretical calculations [23].

## 3. Materials and Methods

### 3.1. Compounds and Activity Data

Bioactive compounds were extracted from the ChEMBL database version 26. Only compounds with reported direct interactions (target relationship type: “D”) with annotation for 35 human aminergic GPCR^16^ targets at the highest confidence level (target confidence score: 9) and exact measurements (“=”) were selected. In addition, only well-defined potency measurements were taken into account (standard type: ‘K_i_’, ‘IC_50_’, ‘EC_50_’, ‘K_b_’, ‘K_d_’, ‘pK_i_’, ‘pIC_50_’, ‘pEC_50_’, ‘pK_b_’, ‘LogK_i_’, or ‘pK_d_’) and standardized in the form of negative decadic logarithm values. Given these criteria, a total of 21,800 unique compounds with activity against 35 targets (44,033 unique measurements in total) were obtained. Compound and activity data were extracted using in-house Python scripts and KNIME [24] protocols with the aid of the Open Eye Toolkit [25].

### 3.2. Fluorinated Compound Sets

The structures of all 21,800 compounds were systematically compared, and if two or more compounds with reported activity against the same target differed only in the position of substituted fluorine atoms, requiring the presence of at least two F analogs, they were combined into an isomeric F-based analog set for SAR analysis (called F_i_SAR set), as illustrated in Table 3. Accordingly, 898 target-based F_i_SAR sets were identified comprising a total of 2163 (1198 unique) fluorinated compounds active against 33 different aminergic GPCRs.

### 3.3. Activity Cliffs

If a pair of fluorinated compounds belonged to the same F_i_SAR set (the only difference is the position of the fluorine atom) and had ΔpPot larger or equal to |1.7| (the difference in potency between the most active and second isomer on a logarithmic scale), it was classified as an AC. A ΔpPot of 1.7 corresponds to a 50-fold increase or decrease in potency, Where an at least 100-fold change in potency is often generally applied as a criterion to define ACs, (i.e., ΔpPot > 2) [26], we applied a lower potency difference threshold because only changes in fluorine positions in a molecule were considered.

### 3.4. Matched Molecular Pairs

For a systematic molecular similarity assessment of different F_i_SAR sets, matched molecular pairs (MMPs) were calculated. MMPs were generated by fragmentation of exocyclic single bonds in compound structures according to Hussain and Rea [27,28], and in this analysis, only transformation size-restricted MMPs [29] after single-cut fragmentation [30] were considered [26]. A transformation size-restricted MMP is an MMP in which the identical part of the two molecules is at least twice the size of the exchanged substructures. In addition, the difference in size between the exchanged substructures is limited to at most eight heavy atoms, and both are not allowed to contain more than 13 heavy atoms [29].

### 3.5. MMP Networks

MMP networks in which nodes represented compounds and edges pairwise MMP relationships were generated using Cytoscape [31]. Compounds can be involved in multiple F_i_SAR sets with activity against different GPCR targets. For MMP network generation, these F_i_SAR sets were combined, as shown in Table 4. The MMP network captured similarity relationships between the resulting 898 F_i_SAR sets. In the MMP network, sets were color-coded according to the aminergic subfamily of GPCRs.

Each of the 898 F_i_SAR sets was represented by a fluorinated compound as a single node, and two nodes were connected by an edge if they formed an MMP. In addition, nodes were shown with a black border if at least one ΔpPot value within the set was larger or equal to |1.7|, thus representing an AC. If one non-F compound was a substructure of another and both formed an MMP (i.e., the MMP transformation involved a hydrogen atom in one compound and a nonhydrogen moiety in the other compound), the edge was colored blue. All remaining edges are colored red.

### 3.6. Computational Workflow Used to Predict the Most Potent Fluorine Derivative

Herein, we developed and evaluated a computational workflow (Figure 8) involving induced-fit docking (IFD), molecular dynamics simulations (MD), and quantum polarized ligand docking (QPLD) combined with energy calculations (applying the Molecular Mechanics Generalized Born Surface Area (MM-GBSA) method). As a dataset to evaluate the proposed workflow, we used F_i_SAR sets for which an AC was available for the crystallized targets.

#### 3.6.1. Induced-Fit Docking (IFD)

The 3-dimensional structures of the ligands were prepared using LigPrep v3.6 [32], and the appropriate ionization states at pH = 7.0 ± 0.5 were assigned using Epik v3.4 [33,34]. Compounds with unknown absolute configurations were docked in both R and S forms. The Protein Preparation Wizard [32] was used to assign the bond orders, appropriate amino acid ionization states, and check for steric clashes for the selected crystal structure (5-HT1a—PDB ID: 7E2Z, 5-HT2a—PDB ID: 6A93, D2—PDB ID: 6CM4, M1—PDB ID: 5CXV, M3—PDB ID: 5ZHP). The receptor grid was generated (OPLS3 force field) by centering the grid box with a size of 8 Å on crystalized ligands. Automated flexible docking of the nonfluorinated compounds was performed using Glide v6.9 [35,36,37] at the SP level.

#### 3.6.2. Molecular Dynamics (MD)

A 100 ns-long molecular dynamics (MD) simulations were performed using Schrödinger Desmond software [38]. Each ligand–receptor complex selected for IFD was immersed into a POPC (309.5 K) membrane bilayer, where the position was calculated using the PPM web server [39]. The system was solvated by water molecules described by the TIP4P potential [40], and the OPLS3e force field [41] was used for all atoms. A total of 0.15 M NaCl was added to mimic the ionic strength inside the cell.

#### 3.6.3. Quantum Polarized Ligand Docking (QPLD)

The grids for the receptors were generated (OPLS3 force field) by centering the grid box on a ligand with a size of 8 Å. Docking of all fluorinated compounds was performed by a quantum-polarized ligand docking (QPLD) [42] procedure involving the QM-derived ligand atomic charges in the protein environment at the BLYP/cc-pVDZ level [43,44]. For each ligand, 5 poses were obtained.

#### 3.6.4. Binding Free Energy Calculations

GBSA (Generalized-Born/Surface Area) was used to calculate the binding free energy based on the ligand–receptor complexes generated by the QPLD procedure. The ligand poses were minimized using the local optimization feature in Prime, the flexible residue distance was set to 6 Å from a ligand pose, and the ligand charges obtained in the QPLD stage were used. The energies of complexes were calculated with the OPLS3e force field and Generalized-Born/Surface Area continuum solvent model. To assess the influence of a given substituent on the binding, ΔΔG was calculated as the difference between the binding free energy (ΔG) of the most active fluorinated compound and its fluorinated analogs.

## 4. Conclusions

Herein, we systematically explored the effect of fluorinated isomers on the activity of aminergic GPCR ligands. A total of 1201 unique fluorinated ligands (2163 in total) were identified for 33 aminergic GPCR targets that had at least two fluorine isomers. Detailed analysis of derived F_i_SAR sets identified a limited number of ACs. However, the results showed that the change in fluorine position could lead to a 1300-fold change in potency, which makes fluorine a “game changer” in the rational design of new and highly potent drugs. Additionally, fluorine atoms can also be used for tuning the selectivity and biological activity toward even very similar targets from one subfamily. Based on the performed analysis, no general rules for the design of the most active fluorine derivative can be deduced. However, we proposed an in silico protocol that takes into account the influence of fluorine atoms on the electron density distribution (i.e., inductive and resonance effects, which are not treated in standard molecular docking). The workflow supports the identification of suitable fluorinated derivatives with the highest biological activity and reduces the cost and time needed for a given derivative synthesis.

The results presented herein demonstrate the importance of fluorine in medicinal chemistry in ligands of aminergic receptors of class A GPCRs, and the proposed computational workflow provides a computational tool for the rational design of new fluorinated drugs.

## Figures and Tables

**Figure 1 molecules-28-00490-f001:**
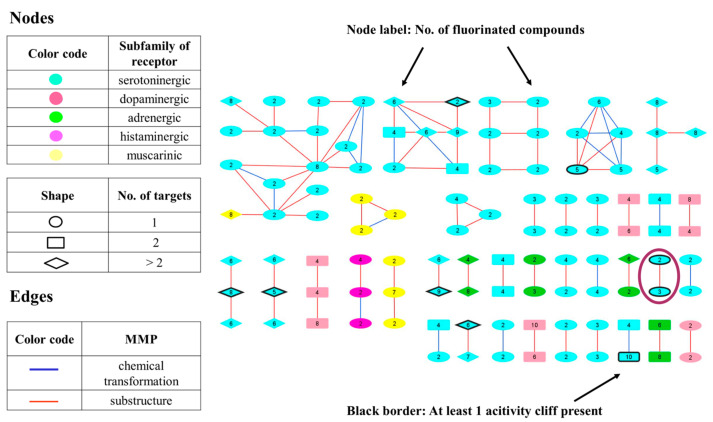
MMP networks. Nodes represent single-, dual-, or multitargeted F_i_SAR sets corresponding to the node shape. The color of the node represents the predominant subfamilies of aminergic receptors of GPCR class A (cyan: serotoninergic, salmon: dopaminergic, green: adrenergic, magenta: histaminic, yellow: muscarinic). Edges between nodes are drawn if they form an MMP—red if they are substructures and blue for other chemical transformations. A thick black border of nodes indicates the presence of at least one AC in the F_i_SAR set. Singletons are not shown; an example presented in Figure 2 is encircled in magenta.

**Figure 2 molecules-28-00490-f002:**
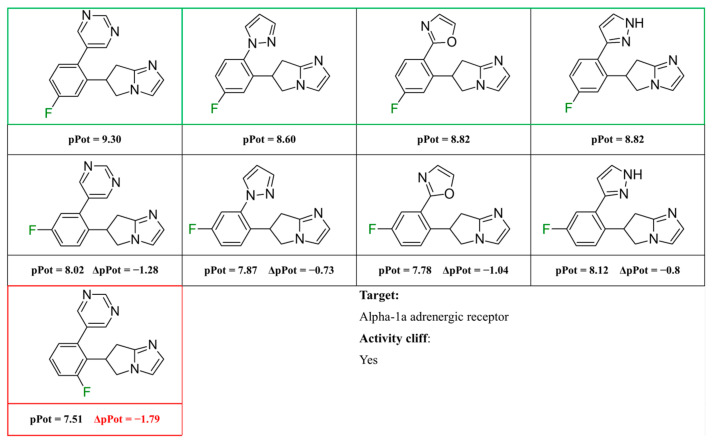
Four exemplary F_i_SAR sets were combined into one multitargeted F_i_SAR set against the alpha-1a adrenergic receptor [15]. For each compound, the pPot and ΔpPot values and corresponding target names are reported below the structures.

**Figure 3 molecules-28-00490-f003:**
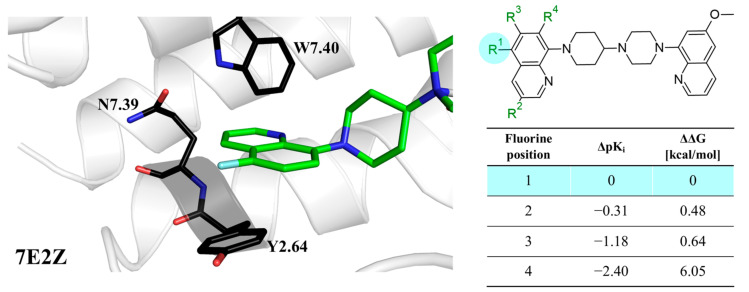
Binding affinities of selected compounds and ΔΔG values (**right**). Representative L–R complex (top scores based on ΔG and pK_i_) of the best ligand with the 5-HT1a receptor. Amino acids crucial for interacting with fluorine in the whole series are shown as sticks (**left**). The binding mode shown contains a compound marked with a cyan circle on the structure and a highlighted row in the table.

**Figure 4 molecules-28-00490-f004:**
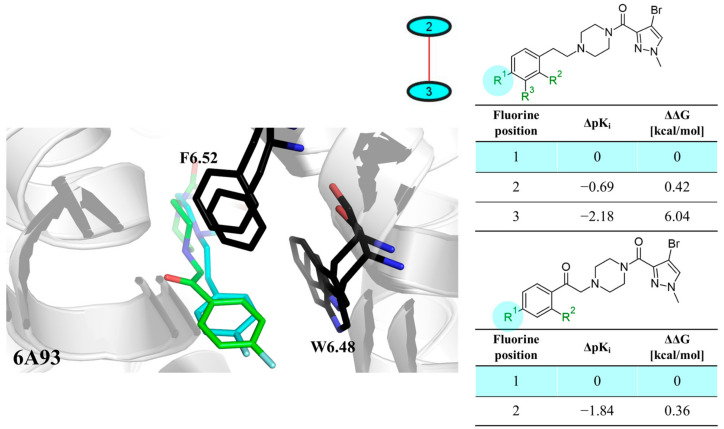
Binding affinities of selected compounds and ΔΔG values (**right**). Representative L–R complex (top scores based on ΔG and pK_i_) of the best ligand with the 5-HT2a receptor. Amino acids crucial for interacting with fluorine in the whole series are shown as sticks (**left**). The binding mode shown contains a compound marked with a cyan circle on the structure and a highlighted row in the table. The presented example is indicated using a magenta rectangle in Figure 2.

**Figure 5 molecules-28-00490-f005:**
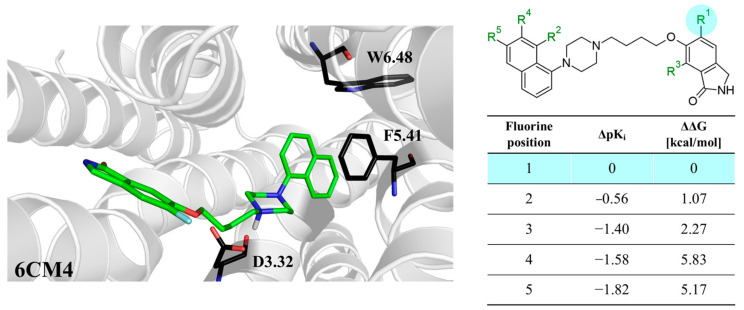
Binding affinities of selected compounds and ΔΔG values (**right**). Representative L–R complex (top scores based on ΔG and pK_i_) of the best ligand within the D2 receptor binding pocket. Amino acids crucial for interacting with fluorine in the whole series and D3.32 are shown as sticks (**left**). The binding mode shown contains a compound marked with a cyan circle on the structure and a highlighted row in the table.

**Figure 6 molecules-28-00490-f006:**
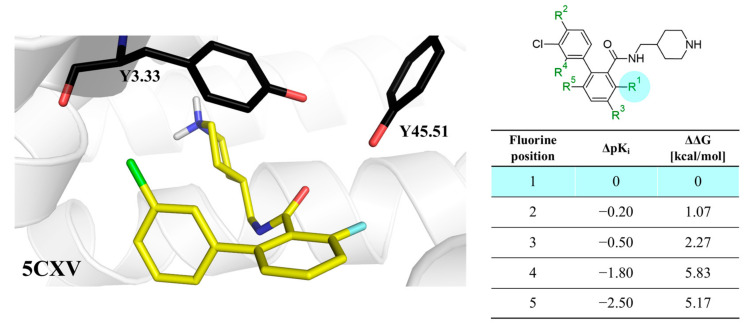
Binding affinities of selected compounds and ΔΔG values (**right**). Representative L–R complex (top scores based on ΔG and pK_i_) of the best ligand with the M1 receptor. Amino acids crucial for interacting with fluorine in the whole series are shown as sticks (**left**). The binding mode shown contains a compound marked with a cyan circle on the structure and a highlighted row in the table.

**Figure 7 molecules-28-00490-f007:**
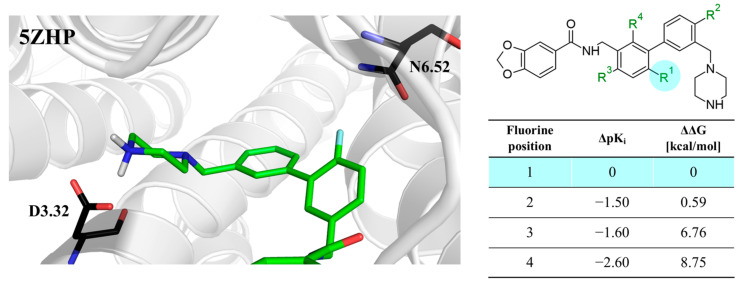
Binding affinities of selected compounds and ΔΔG values (**right**). Representative L–R complex (top scores based on ΔG and pK_i_) of the best ligand within the M3 receptor binding pocket. Amino acids crucial for interacting with fluorine in the whole series are shown as sticks (**left**). The binding mode shown contains a compound marked with a cyan circle on the structure and a highlighted row in the table.

**Figure 8 molecules-28-00490-f008:**
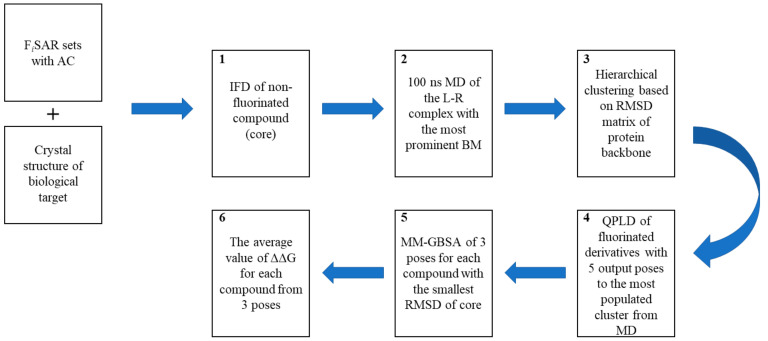
The computational workflow used to predict the most potent fluorine derivative of a compound based on F_i_SAR sets containing an AC and a crystallized biological target. The workflow consisted of IFD of a non-existing nonfluorinated compound (core) (**1**) and 100 ns-long MD simulations (**2**), which were clustered based on the RMSD matrix of the protein backbone (**3**). All fluoro-derivatives were docked to the most frequently observed conformation of the protein using the QPLD algorithm (**4**). For the three conformations of the ligand with the smallest RMSD of the core to nonfluorinated compounds, the MM-GBSA approach was used to calculate the binding energy (ΔG) (**5**). The last step was the calculation of the difference in the interaction energy between the most active compound and subsequent isomers (ΔΔG) (**6**).

**Table 1 molecules-28-00490-t001:** Two exemplary compound sets with activity cliffs are shown. For each compound, the potency and ΔpPot values are displayed. The compounds with |ΔpPot| higher than 1.7 are marked in red.

Target-Based F_i_SAR Set	Fluorinated Compounds			
M_1_	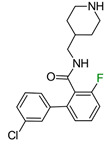	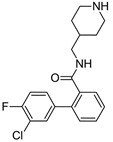	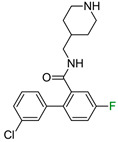	
	**pPot = 8.10**	**pPot = 7.90** **ΔpPot = −0.2**	**pPot = 7.60** **ΔpPot = −0.5**	
	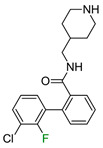	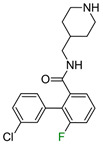		
	**pPot = 6.30** **ΔpPot = −1.8**	**pPot = 5.60** **ΔpPot = −2.5**		
5HT2a	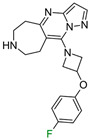	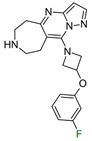	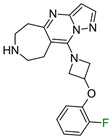	
	**pPot = 8.82**	**pPot = 7.80** **ΔpPot = −1.02**	**pPot = 6.49** **ΔpPot = −2.33**	

**Table 2 molecules-28-00490-t002:** Multitargeted F_i_SAR set. Three exemplary F_i_SAR sets were combined into one multitargeted F_i_SAR set against serotoninergic 5-HT1a, 5-HT2b, and 5-HT1d receptors. For each compound, the pPot, ΔpPot values, and corresponding target names are reported below the structures.

Target-Based F_i_SAR Set	Fluorinated Compounds				Activity Cliffs
	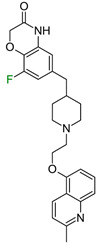	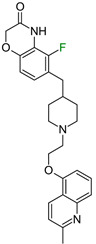	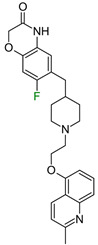	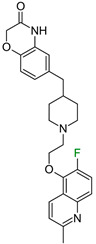	
5HT1b	**pPot = 9.10**	**pPot = 8.60 ** **ΔpPot = −0.5**	**pPot = 7.70** **ΔpPot = −1.4**	**pPot = 7.50** **ΔpPot = −1.6**	NO
5HT1d	**pPot = 9.40**	**pPot = 9.10** **ΔpPot = −0.3**	**pPot = 8.50** **ΔpPot = −0.9**	**pPot = 8.60** **ΔpPot = −0.8**	NO
5HT1a	**pPot = 8.30** **ΔpPot = −1.0**	**pPot = 8.60** **ΔpPot = −0.7**	**pPot = 9.30**	**pPot = 7.30** ** ΔpPot = −2.0 **	YES

**Table 3 molecules-28-00490-t003:** Two exemplary F_i_SAR sets for serotonin receptor 5-HT6.

SET A	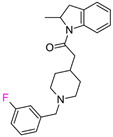	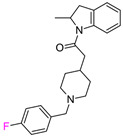	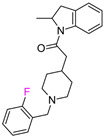	
SET B	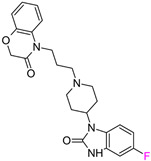	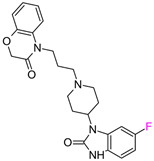	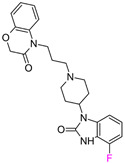	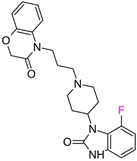

**Table 4 molecules-28-00490-t004:** Two target-based FiSAR sets with ΔpPot values and their individual potency effects for the serotonin 2a (5-HT2a) receptor. The presented example is indicated using a magenta rectangle in Figure 1.

Target-Based F_i_SAR Set	Fluorinated Compounds			Activity Cliffs
5HT2a	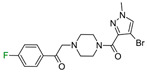	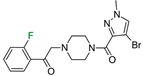		YES
	**pPot = 8.41**	**pPot = 6.56** **ΔpPot = −1.84**		
5HT2a	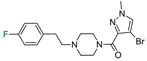	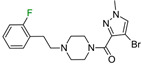	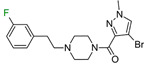	YES
	**pPot = 8.44**	**pPot = 7.74** **ΔpPot = −0.5**	**pPot = 6.27** **ΔpPot = −2.18**	

## Data Availability

The data were obtained from the ChEMBL database (https://www.ebi.ac.uk/chembl**/**, (accessed on 7 February 2022)).

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
