# Peer review of "Isomeric Activity Cliffs—A Case Study for Fluorine Substitution of Aminergic G Protein-Coupled Receptor Ligands"

_molecules, 2023, doi:10.3390/molecules28020490_

Round 1

Reviewer 1 Report

The Manuscript by Pietruś et al, presents an integrated approach that examined fluorine isomers of compounds in the context of activity cliffs (ACs). Overal, the analyses have been well-performed and the Manuscript is also written well. I have only two  concern in this manuscript. 

1) In section 2.1, Authors considered well-defined potency measurements of compounds. I just want to know that did Authors considered all activity scales at the same time or they brought/converted activity values to the same scale (e.g., say IC50). 

2) Prior to docking did Authors validated/benchmarked the docking approach w.r.t. any postive control? 

Reviewer 2 Report

The manuscript titled “Isomeric activity cliffs – a case study for fluorine substitution for aminergic G protein coupled receptors" reports the identification and analysis of 898 Fluor-containing isomeric analog sets for SAR analysis in the ChEMBL database - active FiSAR sets against 33 different aminergic GPCRs comprising a total of 2163 fluorinated compounds (1201 unique). Thus, induced-fit docking, molecular dynamics, quantum polarized ligand docking, and binding free energy calculations based on the Generalized-Born Surface-Area (GBSA) model were performed to improve the discrimination of active from inactive compounds.

The authors explored the effect of fluorinated isomers on the activity of aminergic GPCR ligands and showed that the change in fluorine position could lead to a 1300-fold change in potency, which makes fluorine a "game changer" in the rational design of highly potent new drugs.

Overall the manuscript is rich and interesting; and the paper structure is well-knit and suitable for publication in the journal, after minor revisions. The comments are listed as the following points:

1.          Some corrections should be made (forgotten points to add or others to delete)? to check.

2.          Keywords, line 26, the word "keyword 1" and the numbers 2 to 9 must be deleted.

3.          Line 120, “Figure 2” should be “Table”.

4.          Line 150, “…were prepared using LigPrep v3.6,” Add a reference.

5.          Line 151, “…were assigned using Epik v3.4” Add a reference.

6.          Line 158, “Automated flexible docking of the nonfluorinated compounds was performed using Glide v6.9 at the SP level.” Add a reference.

7.          Line 209, “Figure 4” should be “Table”.

8.          Line 245, “Figure 5” should be “Table”.

Reviewer 3 Report

The paper by Pietrus et al. investigates cases where different ways of introducing fluorine into compounds changes binding to aminergic GPCRs in very different ways. To this end they use literature data and complement simple docking with more elaborate computational tools that use MM-QM couplings in MD simulations. The study is timely and interesting but some parts are not entirely clear to me and need to be better explained.

It took me quite some time to understand figure 6. Part is because the quality of the figure is poor, part because colors are used that some (like me with red/green sufficiency) cannot recognize properly.

more:

    • Numbers in navy-colored shapes are hard to read.
    • Ca. 90 out of 177 nodes are already shown. Why not show everything in its entirety?
    • Maybe it’s standard in the field, but for me it’s not clear what the differences between the edges (red (similarity due to shared substructures) vs blue (similarity due to other chemical transformations)) exactly are.
    • Maybe it would be helpful if one small network is shown in detail (the combined FiSAR sets of the nodes and the shared substructures in the edges). This is anyway half-way done in figures 2, 5 & 7. Maybe these could be combined to a single but more comprehensible figure by showing just one network?
    • At the bottom of the plot, they write “At least 1 activity present”, which likely should be “At least 1 activity cliff present

They found 898 FiSAR set at 2163 compounds and write that there are 2 to 7 compounds in each set. It would be interesting to know how many sets with 2 compounds, how many with 3, etc. exist.

Figure 12 is missing

To create the FiSAR Sets, they used different potency measures (e.g. Ki, IC50, etc). Were there instances with multiple measures for a FiSAR set, and if so, did they all agree on whether an AC was present or not?

One FiSAR Set targets only one receptor. There were 30 FiSAR sets with ACs. Is there overlap between the FiSAR sets if only chemical structures are considered, i.e. are the same compounds leading to ACs in different FiSAR sets? That could maybe be mentioned in 3.3 with the MMP networks. They combine structurally identical compounds into 177 combined FiSAR sets anyway. How many of these sets show ACs?

The MMP and MMP networks are hard to understand. Is it possible to make maybe an example?

If FiSAR sets with the same compounds were combined, how can the network still capture the relationships between 898 FiSAR sets (In 3.3 they write that there are 177 combined sets). Further, they want to make an example for a combined FiSAR set (same compounds but different targets) in figure 2, but I think they show the opposite (same target but slightly different compounds). Line 112: “Compounds can be involved in multiple FiSAR sets with activity against different GPCR targets.”

Line 115: In 2.5 MMP networks, they list the colors for figure 6, which I don’t think is necessary.

Section numbering in methods is off (2.5 appears twice)

Computational part is clearly shown and very well described (in contrast to the MMP part). (But I don’t know enough to really make a comment)

Figure 7:

    • I’m again confused about the definition of a “multitargeted FiSAR set”. In this figure it is “same target and similar (but different) chemical structures”. In the main text (3.3 MMP networks) and Figure 5, however, “multitargeted FiSAR set” seems to describe same chemical structures targeting different receptors. (e.g. Line 238: “… if they shared the same chemical core and differed only in fluorine position”) It’s the same ambiguity as in the methods.
  • The authors compare their computations only with ΔpKi values. Do the ΔΔ G also correspond to the Δ pPot values that were used to identify ACs? Or are all of these cases in which pKi values were used to calculate the pPot values?
  • Usually, a change in pKi by 2 corresponds to a change in ΔΔ G of (very roughly) 5-6 kcal/mol. This is not the case for the second ligand in figure 9, but they don’t really mention it. In general, it is a bit hard to follow in the main text (at least for me it’s not clear which one is the phenethanone and which the phenethyl piperazine linker)
  • Maybe they could mention how many ACs they correctly predicted (or missed) based on their in silico workflow. Since they propose this workflow to score fluorine positions, it might be good to include some summary plot that shows how well the predictions agree with the experimental data.
  • In the conclusion the authors mention about 30 aminergic GPCRs that were targeted. In the abstract and the main text (line 191), they write it were 33.

Since their workflow uses MD simulations with QM/MM couplings of receptor immersed in detergent micelles it is also important to know how much computer times is need to score a single ligand (can this really be done on a larger set of ligands?)
